# Random Cycle Coding: Lossless Compression of Cluster Assignments via Bits-Back Coding

**Daniel Severo**      **Ashish Khisti**      **Alireza Makhzani**

University of Toronto
Department of Electrical and Computer Engineering
`{d.severo@mail, akhisti@, a.makhzani@}.utoronto.ca`

## Abstract

We present an optimal method for encoding cluster assignments of arbitrary data sets. Our method, *Random Cycle Coding* (RCC), encodes data sequentially and sends assignment information as cycles of the permutation defined by the order of encoded elements. RCC does not require any training and its worst-case complexity scales quasi-linearly with the size of the largest cluster. We characterize the achievable bit rates as a function of cluster sizes and number of elements, showing RCC consistently outperforms previous methods while requiring less compute and memory resources. Experiments show RCC can save up to $2$ bytes per element when applied to vector databases, and removes the need for assigning integer ids to identify vectors, translating to savings of up to $70\%$ in vector database systems for similarity search applications.

## 1   Introduction

A *clustering* is a collection of pairwise disjoint sets, called *clusters*, used throughout science and engineering to group data under context-specific criteria. A clustering can be decomposed conceptually into two parts of differing nature. The *data set*, created by the set union of all clusters, and the *assignments*, indicating which elements belong to which cluster. This work is concerned with the *lossless* communication and storage of the assignment information, for arbitrary data sets, from an information theoretic and algorithmic viewpoint.

Communicating clusters appears as a fundamental problem in modern vector similarity databases such as FAISS [Johnson et al., 2019]. FAISS is a database designed to store vectors of large dimensionality, usually representing pre-trained embeddings, for similarity search. Given a query vector, FAISS returns a set of the $k$-nearest neighbors [Lloyd, 1982] available in the database under some pre-defined distance metric (usually the L2 distance). Returning the exact set requires an exhaustive search over the entire database for each query vector which quickly becomes intractable in practice. FAISS can instead return an approximate solution by performing a two-stage search on a coarse and fine grained set of database vectors. The database undergoes a training phase where vectors are clustered into sets and assigned a representative (i.e., a centroid). FAISS first selects the $k'$-nearest clusters, $k' < k$, based on the distance of the query to the centroids, and then performs an exhaustive search within them to return the approximate $k$-nearest neighbors.

The cluster assignments must be stored to enable cluster-based approximate searching. In contrast to a class, a cluster is distinguishable only by the elements it contains, and is void of any labelling. However, cluster assignments are often stored alongside the data set in the form of artificially generated labels. Lossy compression techniques for storing the vectors themselves is an active area of research [Chen et al., 2010, Martinez et al., 2016, Babenko and Lempitsky, 2014, Jegou et al., 2010, Huijben et al., 2024]. In this literature, the number of bits used to store the vector embedding ranges

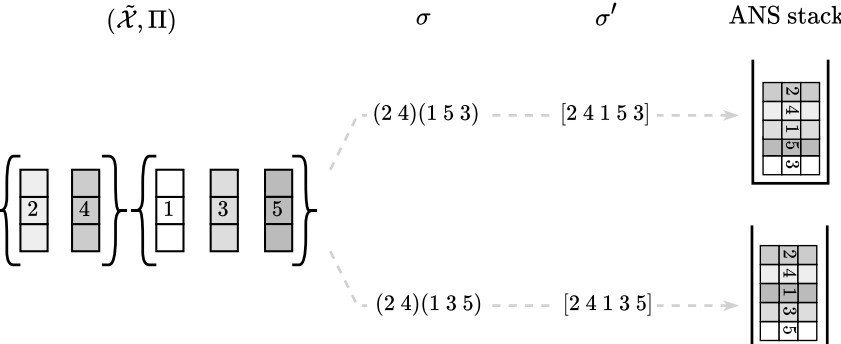

Figure 1: High-level description of our method, Random Cycle Coding (RCC). RCC encodes the clustering $(\tilde{\mathcal{X}}, \Pi)$ as cycles in the permutation $\sigma$ induced by the relative ordering of objects. **Left:** Indices represent the rankings of objects in $\tilde{\mathcal{X}}$ according to the total ordering of $\mathcal{X}$. **Middle:** One of two permutations will be randomly selected with bits-back coding to represent the clustering ($\sigma$, shown in cycle-notation). Then, Foata's Bijection is applied to yield $\sigma'$, shown in line-notation, which is encoded into the ANS stack. **Right:** The final ANS stack containing $\sigma'$ in line-notation.

from 4 to 16 bytes, while ids are typically stored as 8-byte integers. Therefore, labelling, for the sake of clustering, can represent the majority of bits spent for communication and storage in a typical use case, and will become the dominating factor as the performance of these compression algorithms improves.

In this work we show how to communicate and store cluster assignments without creating artificial labels, providing substantial storage savings for vector similarity search applications. Assignments are implicitly represented by a cycle of a permutation defined by the order between encoded elements. Our method, *Random Cycle Coding* (RCC), uses bits-back coding [Townsend et al., 2019] to pick the order in which data points are encoded. The choice of orderings is restricted to the set of permutations having disjoint cycles with elements equal to some cluster. RCC is optimal as it achieves the Shannon bound [Cover, 1999] in bit savings. The worst-case computational complexity of RCC is quasi-linear in the largest cluster size and requires no training or machine learning techniques.

An overview on lossless compression, permutations, and bits-back coding is given in Section 2. Our method is presented and analyzed in Section 3. To the best of our knowledge, no current method exists to optimally store cluster assignments. In Section 4 we provide two strong baselines based on the work of Severo et al. [2023] for compressing multisets. Section 5 showcases RCC on real-world databases from a well known vector database called FAISS [Johnson et al., 2019], achieving savings of up to 70% in the best case.

## 2 Background

### 2.1 Lossless Compression

Lossless compression aims to find a code $C : \mathcal{X} \mapsto \{0, 1\}^\star$ for an infinite i.i.d. sequence of elements $X^{(i)} \sim P_X$ that can be decoded with perfect fidelity. Lossless decoding requires restricting $C$ such that the extended code $C(X^{(1)})C(X^{(2)}) \ldots$, created via concatenation, is uniquely decodable. It is known that $\mathbb{E}_{X \sim P_X}[C(X)] \geq H(P_X)$ for any uniquely decodable code, and any code with average-length close to $H(P_X)$ must obey $C(x) \approx -\log P_X(x)$ [Shannon, 1948, Cover, 1999]. It is possible to construct $C$ using entropy coders such as Asymmetric Numeral Systems [Duda, 2009], Arithmetic Coding [Witten et al., 1987], or Huffman Codes [Huffman, 1952].

A common way of designing efficient codes is to guarantee the code-word $C(x)$ carries some meaningful semantics for $x$ that allows for an efficient implementation. Semantic codes can be constructed by introducing an intermediate sequence of random variables $Z^n$, acting as a proxy for $X$[1], which is entropy coded autoregressively in $n$ steps. The code-word for $x$ is the binary

---

[1]From here on we drop the superscript on $X^{(i)}$ and focus on a single $X$, without loss of generality.

representation of the final state of the chosen entropy coder (e.g., ANS [Duda, 2009]). Unfortunately, semantic entropy coding can lead to sub-optimal performance due to the non-uniqueness of the mapping between $\mathcal{X}$ and $\mathcal{Z}^n$, limiting their practical use in applications where this redundancy is large. This is the case for large structured data types such as clusters of high-dimensional embeddings.

Asymmetric Numeral Systems (ANS) [Duda, 2009] is an entropy coder that stores data to an integer state $s \in \mathbb{N}$ using the PMF and CDF of some probability model over the data. When data is encoded the integer state increases by approximately the information content under the model (i.e., the negative log-likelihood), which equals the entropy of the source on average. The final value of the integer state is then serialized to disk using approximately $\log s$ bits. ANS operates on $s$ in a stack-like fashion. Symbols are decoded in reverse order in which they where encoded. Due to this stack-like nature ANS can be used as an invertible sampler, by initializing the stack to a random integer and performing a decode operation. Under mild initialization conditions for $s$ (see Townsend et al. [2019], Severo et al. [2023]), the decoded sample will be distributed according to the probability model used for decoding. Decoding reduces the number of bits required to represent the stack by the information content of the sampled symbol. The randomly initialized stack can be recovered by encoding the sampled symbol back into the stack using the same distribution.

## 2.2 Bits-back Coding with ANS

Bits-back Coding [Frey and Hinton, 1996, Townsend et al., 2019] is an entropy coding method for latent variable models $P_{X,Z}$. The bit-rate achieved is equal to the cross-entropy $\mathbb{E}_X\left[-\log P_X(X)\right]$, where the expectation is taken with respect to the true data distribution of $X$, despite not having direct access to the marginal $P_X$ needed for entropy coding. Given the posterior $P_{Z\,|\,X}$, prior $P_Z$, and conditional likelihood $P_{X\,|\,Z}$ of the model, bits-back coding using ANS to perform invertible sampling from $P_{Z\,|\,X}$ to obtain $Z$. Then, $X$ is encoded with $P_{X\,|\,Z}$, conditioned on the sampled $Z$. Finally, $Z$ is encoded with the prior $P_Z$. Decoding from the stack reduces the ANS integer state by $-\log P_{Z\,|\,X}$ while encoding increases it by $-\log P_Z P_{X\,|\,Z}$, resulting in a net increase equal to the cross-entropy. An approximate posterior can be used when the exact posterior $P_{Z\,|\,X}$ is not available, resulting in the net increase being equal to the Negative Evidence Lower-Bound (NELBO) [Townsend et al., 2019].

## 2.3 Random Order Coding (ROC)

ROC [Severo et al., 2023] is an algorithm for losslessly compressing multisets[2]. A sequence can be seen as a multiset, representing the frequency count of symbols, together with a permutation defining the ordering. ROC uses bits-back coding with the ordering as a latent variable $Z$, and the multiset as the observation $X$, together with an exact posterior $Q_{Z\,|\,X} = P_{Z\,|\,X}$. The exactness of the posterior implies the number of bits used by ROC to encode the multiset is equal to the cross-entropy of the multiset with respect to the true data distribution.

ROC applies a similar procedure to our method to select a random element in the multiset which is then encoded with a symbol codec using ANS. For a multiset with $k$ *unique* elements, each appearing $n_i$ times, ROC saves exactly $-\log P_{Z\,|\,X} = \log\binom{n}{n_1,\ldots,n_k} \leq \log(n!)$ bits, where the quantity in parentheses is the multinomial coefficient.

## 2.4 Permutations and Cycles

A *permutation*, in this work, is a bijective function $\sigma : [n] \mapsto [n]$ used to define arrangements of elements from arbitrary sets. Permutations are usually expressed in one-line notation $\sigma = [i_1, i_2, \ldots, i_n]$ where $\sigma(j) = i_j$. The *symmetric group* $\mathcal{S}_n$ on $n$ elements is the set of all permutations of $[n]$.

A *cycle* $(c_1 \ \ldots \ c_k)$, of a permutation $\sigma$, is the sequence constructed from the repeated application of $\sigma$, to some element $c_1 \in [n]$, until $c_1$ is recovered, i.e., $c_{k+1} = c_1$, $c_i = \sigma(c_{i-1})$, for $i \geq 2$.

*Example* 2.1. The cycles of $\sigma = [3, 1, 2, 5, 4]$ are shown below.
A permutation can be represented by its cycles with the following procedure. Pick any element in $[n]$ and compute its cycle by applying $\sigma$ successively. Next, choose another element in $[n]$, that did not show up in any of the previously computed cycles, and compute its cycle. Repeat this procedure

---

[2]Multisets are sets that allow repetition of elements but have no ordering between elements.

$$\begin{array}{cccccc}
c_1 & 1 & 2 & 3 & 4 & 5 \\
\text{cycle} & (1\,3\,2) & (2\,1\,3) & (3\,2\,1) & (4\,5) & (5\,4)
\end{array}$$

until all elements appear in exactly one cycle. Concatenate cycles to form the representation. Every permutation has a unique representation through the concatenation of disjoint cycles. For example,

$$\sigma = [3, 1, 2, 5, 4] = (1\,3\,2)(4\,5), \tag{1}$$

where the right-hand side is called the cycle notation of $\sigma$.

**Lemma 2.2** (Foata's Bijection [Foata, 1968])**.** The following sequence of operations defines a bijection between permutations on $n$ elements. Write the permutation in disjoint cycle notation such that the smallest element of each cycle appears first within the cycle. Order the cycles in decreasing order based on the first/smallest element in each cycle. Remove all parenthesis to form the one-line notation of the output permutation.

*Example* 2.3. Applying the steps in the construction of Foata's bijection to $[3, 1, 2, 5, 4] = (3\,2\,1)(5\,4)$ yields $(1\,3\,2)(4\,5) \mapsto (4\,5)(1\,3\,2) \mapsto [4, 5, 1, 3, 2]$. Cycles can be recovered by scanning from left to right and keeping track of the smallest value.

## 3 Method

In this section we develop our method, *Random Cycle Coding* (RCC), which encodes clusters of data points as disjoint permutation cycles. In what follows, we first describe the coding procedure and then show the model resulting from our procedure assigns likelihood proportional to the product of cluster sizes.

Let $X^n = (X_1, \ldots, X_n)$ be a sequence of random variables $X_i$ with common, but arbitrary, alphabet $\mathcal{X}$. Throughout we assume that a *total ordering* can be defined for $\mathcal{X}$, i.e., elements of the set can be compared and ranked/sorted according to some predefined criteria (e.g., lexicographical ordering). We assume no repeats happen in the sequence. This is motivated by applications where elements are high-dimensional vectors such as embeddings or images where repeats are unlikely to happen.

We are interested in the setting where the elements of $X^n$ are grouped into pair-wise disjoint sets known as *clusters*. Clusters can be represented by a collection of indicator random variables $\Pi = (\Pi_{ij})_{1 < i < j < n}$ where $\Pi_{ij} = 1$ if $X_i, X_j$ are in the same cluster and $\Pi_{ij} = 0$ otherwise. Conditioned on the sequence $X^n = x^n$ the clustering $\Pi$ defines a partition of the data set $\tilde{\mathcal{X}} = \{x_1, \ldots, x_n\}$. The size of the alphabet of $\Pi$ is equal to the number of ways in which $\tilde{\mathcal{X}}$ can be partitioned; known as the $n$-th Bell number [Bell, 1934]. The order between elements in a cluster is irrelevant and clusters are void of labels.

The objective is to design a lossless code for the assignments $\Pi$ that can be used alongside any codec for the data set $\tilde{\mathcal{X}}$. Our strategy will be to send the elements of $\tilde{\mathcal{X}}$ in a particular ordering such that it implicitly encodes the clustering information.

We associate a permutation $\sigma_{x^n}$ to each of the possible $n!$ orderings of $\tilde{\mathcal{X}}$ based on sorting. Let $\mathcal{S}_n(\tilde{\mathcal{X}})$ be the set of all possible orderings of $\tilde{\mathcal{X}}$, and $s^n$ a reference sequence created by sorting the elements in $\tilde{\mathcal{X}}$ according to the total ordering of $\mathcal{X}$,

$$\mathcal{S}_n(\tilde{\mathcal{X}}) = \{x^n : x_i \in \tilde{\mathcal{X}} \text{ and } x_i \neq x_j \text{ for } i \neq j\}, \tag{2}$$

$$s^n \in \mathcal{S}_n(\tilde{\mathcal{X}}) \text{ s.t. } s_1 < s_2 < \cdots < s_n. \tag{3}$$

For any $x^n \in \mathcal{S}_n(\tilde{\mathcal{X}})$, the induced permutation $\sigma_{x^n}$ is defined as that which permutes the elements of the reference $s^n$ such that $x^n$ is obtained. Under this definition the permutation can also be constructed by directly substituting $x_i$ for its *ranking* in $\tilde{\mathcal{X}}$.

The induced permutations allow us to redefine $\Pi$ as a *random equivalence class* taking on values in the *quotient set*, $\mathcal{S}_n(\tilde{\mathcal{X}})/\sim$, of the equivalence relation described next. Note the quotient set is finite even if $\mathcal{X}$ is uncountable as only the relative ordering between elements is needed to define the equivalence relation.

**Definition 3.1** (Cycle-Cluster-Equivalence)**.** Two sequences in $\mathcal{S}_n(\tilde{\mathcal{X}})$ are *equivalent* ($\sim$) if the disjoint cycles of their induced permutations contain the same elements. Given a sequence, two

elements of $\tilde{\mathcal{X}}$ are in the same cluster if their rankings appear in the same disjoint cycle of the induced permutation.

*Example* 3.2. Let $\mathcal{X}$ be the set of even integers under the usual ordering for natural numbers. Sequences $x^n = (4, 6, 2, 8)$ and $z^n = (6, 2, 4, 8)$ induce permutations $\sigma_{x^n} = [2, 3, 1, 4]$ and $\sigma_{z^n} = [3, 1, 2, 4]$. The sequences are equivalent as the disjoint cycles of the induced permutations contain the same elements: $\sigma_{x^n} = (4)(1\ 2\ 3)$, $\sigma_{z^n} = (4)(1\ 3\ 2)$. For both sequences, elements 2, 4, and 6 are in the same cluster, while 8 is in a cluster of its own. The partition, viewed as an equivalence class, is equal to $\Pi = \{x^n, z^n\}$, as there are no other permutations over $\tilde{\mathcal{X}}$ that are equivalent to the two shown.

Given $(\tilde{\mathcal{X}}, \Pi)$, Random Cycle Coding (RCC) uses bits-back coding to send the elements of $\tilde{\mathcal{X}}$ in an ordering which corresponds to a sequence in $\Pi$. The receiver decodes the elements $X_i$, in the order sent, and recovers $\Pi$ by computing the cycles of the induced permutation. The clustering information $\Pi$ is communicated via the cycles of the permutation. See Figure 1 for a high-level description.

Every sequence in $\Pi$ maps to the same clustering over $\tilde{\mathcal{X}}$. The log of the number of elements in the equivalence class equals the redundancy discussed in Section 2.1, which is known to be

$$\log|\Pi| = \sum_{i=1}^{\#\text{cycles}} \log((n_i - 1)!), \tag{4}$$

where $n_i$ is the number of elements in the $i$-th cycle of the induced permutation. An optimal bits-back method must remove exactly $\log|\Pi|$ bits from the stack during encoding. RCC achieves these savings by selecting an element from $\tilde{\mathcal{X}}$, using an ANS decode operation, and then encoding it onto the stack. Interleaving decoding/sampling and encoding avoids the *initial bits* issue [Townsend et al., 2019] resulting from the initially empty ANS stack.

Random Order Coding [Severo et al., 2023] (ROC) performs a similar procedure for multiset compression where elements are also sampled without replacement from $\tilde{\mathcal{X}}$. However, there, the equivalence classes consist of all permutations over $\tilde{\mathcal{X}}$, and therefore sampling can be done by picking any element from $\tilde{\mathcal{X}}$ uniformly at random. RCC requires sampling without replacement from $\tilde{\mathcal{X}}$ non-uniformly such that the resulting permutation has a desired cycle structure. To do so, we define the following procedure, reminiscent of Foata's Bijection [Foata, 1968].

**Definition 3.3** (Foata's Canonicalization). The following steps map all sequences in the same equivalence class, $x^n \in \Pi$, to the same *canonical* sequence $c^n \in \Pi$. First, write the permutation in disjoint cycle notation and sort the elements within each cycle, in ascending order, yielding a new permutation. Next, sort the cycles, based on the first (i.e., smallest) element, in descending order.

*Example* 3.4. The set composed of permutations $\sigma = (3\ 1)(5\ 2\ 4)$ and $\pi = (3\ 1)(2\ 5\ 4)$ is an equivalence class. Applying Foata's Canonicalization to either $\sigma$ or $\pi$ yields $(2\ 4\ 5)(1\ 3)$, which is equal to $\sigma$.

**Algorithm** RCC encodes a permutation using the procedure outlined in Algorithm 1. The elements of $\tilde{\mathcal{X}}$ are inserted into lists according to their clusterings. The clustering is canonicalized according to Definition 3.3. The encoder starts from the last, i.e., right-most, list. The list is encoded as a set using ROC, with the exception of the smallest element, which is held-out and encoded last. This procedure repeats until all lists are encoded. Decoding is shown in Algorithm 2. During decoding the first element is known to be the smallest in its cycle. The decoder then decodes the remaining cycle elements using ROC, and stops when it sees an element smaller than the current smallest element. This marks the start of a new cycle and repeats until all elements are recovered. Python code for encoding and decoding are given in Appendix A.

**Savings** Encoding the smallest value last guarantees that the cycle structure is maintained. Permuting the remaining elements in the cycle spans all permutations in $\Pi$. For the $i$-th cluster with $n_i$ elements the savings from encoding $n_i - 1$ elements with ROC is equal to $\log((n_i - 1)!)$. The total savings is equal to (4), implying RCC saves $\log|\Pi|$.

**Implied Probability Model** The probability model $Q_{\Pi \mid \tilde{\mathcal{X}}}$ is indirectly defined by the savings achieved by RCC. The set of elements and clustering assignments $(\tilde{\mathcal{X}}, \Pi)$ are encoded via a sequence $x^n \in \Pi$. We can assume some lossless source code is used for the data points, requiring

---
**Algorithm 1:** Pseudo-code for encoding with RCC.
---
**Inputs:** (1) Clustering as a set of sets $\{\{x_1^1, \ldots, x_{n_1}^1\}, \{x_1^2, \ldots, x_{n_2}^2\}, \ldots, \{x_1^\ell, \ldots, x_{n_\ell}^\ell\}\}$; (2) Initial ANS state; (3) Symbol codec

1 Sort the clustering into a list of lists, with elements $y_i^c$, according to Definition 3.3, such that $(y_1^c, \ldots, y_{n_c}^c)$ is an increasing sequence, and $(y_1^1, \ldots, y_1^\ell)$ is decreasing.

 **for** $c = \ell, \ldots, 1$ **do**
2 $\quad$ Encode $\left[y_2^c, \ldots, y_{n_c}^c\right]$ with ROC using the given symbol codec
3 $\quad$ Encode $y_1^c$ with symbol codec
 **end**
 **return** Final ANS state
---

---
**Algorithm 2:** Pseudo-code for decoding with RCC.
---
**Inputs:**
- Total number of elements $n$
- Final ANS state, constructed from Figure 4
- Symbol Codec

Initialize $\Pi = [\,], c = 0$
**while** *total number of elements in $\Pi$ is less than $n$* **do**
$\quad$ Decode $y_1^c$ with the symbol codec
$\quad$ Decode elements $y_i^c$ with ROC until an element smaller than $y_1^c$ is seen
$\quad$ Add all decoded elements to $\Pi$ as a list $[y_1^c, \ldots, y_{n_c}^c]$
$\quad$ Increment $c$
**end**
**return** $\Pi$, Initial ANS state
---

$-\log Q_{X^n}(x^n)$ bits to encode the sequence. The cost of encoding the dataset and cluster assignments equals the cost of encoding a sequence minus the discount given by bits-back,

$$-\log Q_{\tilde{\mathcal{X}}, \Pi}(\tilde{\mathcal{X}}, \Pi) = -\log Q_{X^n}(x^n) - \log|\Pi|. \tag{5}$$

From Severo et al. [2023] we know the cost of encoding the set $\tilde{\mathcal{X}}$ is that of the sequence minus the cost of communicating an ordering,

$$-\log Q_{\tilde{\mathcal{X}}}(\tilde{\mathcal{X}}) = -\log Q_{X^n}(x^n) - \log(n!). \tag{6}$$

From this, we can write,

$$-\log Q_{\Pi \mid \tilde{\mathcal{X}}}(\Pi \mid \tilde{\mathcal{X}}) = \log Q_{\tilde{\mathcal{X}}}(\tilde{\mathcal{X}}) - \log Q_{X^n}(x^n) - \log|\Pi| = \log(n!) - \log|\Pi|. \tag{7}$$

The implied probability model only depends on the cluster sizes, and assigns higher likelihood when there are few clusters with many elements,

$$Q_{\Pi \mid \tilde{\mathcal{X}}}(\Pi \mid \tilde{\mathcal{X}}) = \frac{\prod_{i=1}^k (n_i - 1)!}{n!}, \tag{8}$$

where $n_i$ is the size of the $i$-th cluster, and $k$ the total number of clusters.

**Complexity** The complexity of RCC will vary significantly according to the number of clusters and elements. Initializing RCC requires sorting elements within each cluster, which can be done in parallel, followed by a sort across clusters. ROC is used as a sub-routine and has both worst- and average-case complexities equal to $\Omega(n_i \log n_i)$ for encoding and decoding the $i$-th cluster. The total computational complexity of our method, RCC, adapts to the size of the equivalence class: $\Omega\left(\sum_i n_i \log n_i\right) = \Omega(\log|\Pi|)$. When only one permutation can represent the cluster assignments, i.e., $n = k$, implying $\log|\Pi| = 0$, then RCC has the same complexity as compressing a sequence: $\Omega(n)$.

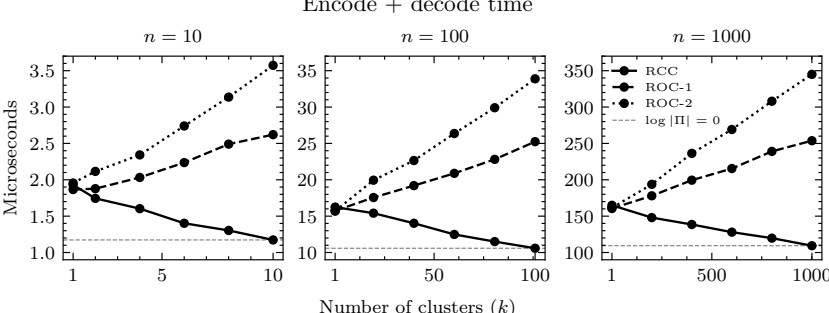

Figure 2: Median encoding plus decoding times, across 100 runs, for Random Order Coding (ROC) [Severo et al., 2023] and our method Random Cycle Coding (RCC). The number of elements $n$ increases from left-to-right across plots. Clusters are fixed to have roughly the same size, $n/k$, mirroring vector database applications discussed in Section 5.3. Reported times are representative of the amount of compute needed to sample the permutation in the bits-back step as data vectors are encoded with ANS using a uniform distribution.

## 4 Related Work

To the best of our knowledge there is no other method which can perform optimal lossless compression of clustered high-dimensional data.

Severo et al. [2023] presented a method to optimally compress multisets of elements drawn from arbitrary sets called Random Order Coding (ROC). ROC can compress clusterings by viewing them as sets of clusters, but is sub-optimal as it requires encoding the cluster sizes. We compare RCC against the following two variants of ROC next and provide experiments in Section 5.

**ROC-1** The cluster sizes are communicated with a uniform distribution of varying precision and clusters are then encoded into a common ANS state. Each cluster contributes $\log(n_i!)$ to the bits-back savings of ROC, resulting in a reduction in bit-rate of

$$\Delta_{\text{ROC-1}} = \sum_{i=1}^{k} \left(\log(n_i!) - \log(n - N_i)\right) \tag{9}$$

$$= \sum_{i=1}^{k} \log\left(\frac{n_i}{n - N_i}\right) + \log|\Pi| \tag{10}$$

$$\leq \log|\Pi|, \tag{11}$$

where $k$ is the number of clusters, $N_i = \sum_{j=1}^{i-1} n_j$ counts the number of encoded elements before step $i$, and $\log(n - N_i)$ is the cost of encoding the size of the $i$-th cluster. The gap to optimality increases with the number of clusters, while RCC is always optimal as it achieves $\log|\Pi|$ for any configuration of elements and clusters.

**ROC-2** This variant views the clusterings as a set of sets. The cluster sizes are communicated as in ROC-1. However, an extra bits-back step is done to randomly select the ordering in which the $k$ clusters are compressed, resulting in further savings. The complexity of this step scales quasi-linearly with the number of clusters, $\Omega(k \log k)$, and requires sending the number of clusters ($\log(n)$ bits), which is also the size of the outer set. The total reduction in bit-rate is

$$\Delta_{\text{ROC-2}} = \Delta_{\text{ROC-1}} + \log(k!) - \log(n). \tag{12}$$

This method achieves a better rate than ROC-1, but can require significantly more compute and memory resources due to the extra bits-back step to select clusters compared to both ROC-1 and RCC. Conditioned on knowing the cluster sizes, ROC-2 compresses each cluster independently. Intuitively, the method does not take into account that clusters are pairwise disjoint and their union equals the interval of integers from 1 to $n$, which explains why it achieves a sub-optimal rate savings.

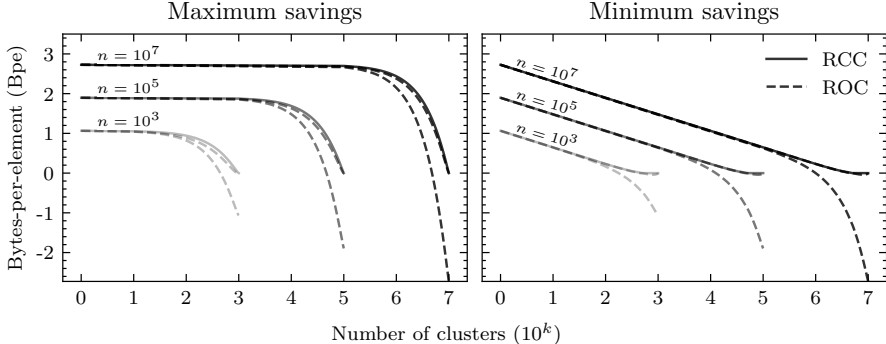

Figure 3: Maximum (left) and minimum (right) byte savings per element as a function of the number of clusters and elements. Savings are maximized when one cluster contains most elements and all others are singletons. The minimum is achieved if clusters have roughly the same number of elements. Two variants of Random Order Coding (ROC) [Severo et al., 2023] are shown (see Section 4) with dashed lines. Random Cycle Coding (RCC) achieves higher savings than both variants of ROC while requiring less memory and computational resources.

## 5 Experiments

### 5.1 Minimum and maximum achievable savings

In applications targeted by our method (see Section 5.3) the cluster size is set according to some budget and elements are allocated into clusters via a training procedure. For a fixed number of elements ($n$) and clusters ($k$) the savings for ROC-1, ROC-2, and RCC will depend only on the cluster sizes ($n_1, \ldots, n_k$). We empirically analyzed the minimum and maximum possible savings as a function of these quantities. Results are shown in Figure 3.

The term dominating the bit savings of all algorithms is the product of the factorials of cluster sizes, $\prod_i n_i!$, constrained to $\sum_i n_i = n$ and $n_i \geq 1$. The maximum is achieved when $n - k$ elements fall into one cluster, $n_j = n - k + 1$, and all others are singletons: $n_i = 1$ for $i \neq j$. Savings are minimized when all clusters have roughly the same size: $n_i = (n \div k) + \mathbf{1}\{i \leq n \bmod k\}$[3].

All methods provide similar savings when $k \ll n$. RCC has better maximal and minimal savings than both ROC-1 and ROC-2 in all settings considered. The need to encode cluster sizes, without exploiting the randomness of cluster orders as in ROC-2, results in ROC-1 achieving *negative* savings when the number of clusters $k$ is large. RCC savings converge to 0 bits as the number of clusters approaches the number of elements, as expected. As $k$ approaches $n$, ROC-2 also suffers from negative savings, but the values are negligible compared to those of ROC-1.

### 5.2 Encoding and decoding times

Figure 2 shows the total encoding plus decoding time as a function of number of elements and clusters. RCC outperforms both variants of ROC in terms of wall-time by a wide margin, while achieving the optimal savings. RCC is compute adaptive and requires the same amount of time to encode a sequence when $\log|\Pi| = 0$. The compute required for ROC variants increases with the number of clusters, correlating negatively with $\log|\Pi|$. ROC-2 is slower than ROC-1 due to the extra bits-back step needed to select clusters with ANS.

### 5.3 Inverted-lists of Vector Databases (FAISS)

We experimented applying ROC and RCC to FAISS [Johnson et al., 2019] databases of varying size. Results are shown in Table 1. Scalar quantization [Cover, 1999, Lloyd, 1982] was used to partition the set of vectors into disjoint clusters. This results in clusters of approximately the same number of

---

[3] $\div$ represents integer division, $n \bmod k$ is the remainder, and $\mathbf{1}\{\}$ is the indicator function that evaluates to 1 if the expression is true.

| Dataset | # Elements | # Clusters | Savings | | | | | |
|---|---|---|---|---|---|---|---|---|
| | | | Max. | Min. | $\frac{1}{8n}\log|\Pi|$ | RCC | ROC-2 | ROC-1 |
| SIFT1M | 1,000,000 | 500 | 2.31 | 1.19 | 1.20 | **0.00** | **0.00** | 0.04 |
| | | 1000 | 2.31 | 1.06 | 1.08 | **0.00** | **0.00** | 0.10 |
| | | 4,973 | 2.30 | 0.77 | 0.81 | **0.00** | 0.04 | 0.86 |
| | | 9,821 | 2.29 | 0.65 | 0.71 | **0.00** | 0.12 | 2.17 |
| | | 46,293 | 2.20 | 0.37 | 0.48 | **0.00** | 1.28 | 18.28 |
| | | 95,284 | 2.07 | 0.24 | 0.30 | **0.00** | 2.07 | 61.43 |
| BigANN | 1,000,000 | 1,000 | 2.31 | 1.06 | 1.07 | **0.00** | **0.00** | 0.10 |
| | | 10,000 | 2.29 | 0.65 | 0.66 | **0.00** | 0.01 | 2.27 |
| | | 99,946 | 2.06 | 0.23 | 0.25 | **0.00** | 0.79 | 76.28 |
| | 10,000,000 | 1,000 | 2.73 | 1.48 | 1.49 | **0.00** | **0.00** | 0.01 |
| | | 10,000 | 2.72 | 1.06 | 1.07 | **0.00** | **0.00** | 0.14 |
| | | 99,998 | 2.70 | 0.65 | 0.66 | **0.00** | 0.01 | 2.89 |

Table 1: Byte savings, per element, from compressing SIFT1M [Jegou et al., 2010] and BigANN [Jégou et al., 2011] as a function of number of elements and clusters. Values in columns RCC, ROC-2, and ROC-1, indicate the gap, in percentage (lower is better), to the optimal savings in bytes-per-element, in column $\frac{1}{8n}\log|\Pi|$. A value of $0.00$ indicates the method achieves the maximum bit savings shown in column $\frac{1}{8n}\log|\Pi|$. Columns Max. and Min. show the theoretical maximum and minimum savings as discussed in Section 5.1.

| $n$ | $\frac{1}{8n}\log|\Pi|$ | % Savings | | | | | |
|---|---|---|---|---|---|---|---|
| | | Sequential ids | | | External ids | | |
| | | 4 | 8 | 16 | 4 | 8 | 16 |
| 1M | 1.06 | 54.8 | 33.9 | 19.2 | 8.9 | 6.7 | 4.4 |
| 10M | 1.27 | 60.5 | 38.3 | 22.1 | 10.6 | 8.0 | 5.3 |
| 100M | 1.48 | 65.6 | 42.4 | 24.9 | 12.3 | 9.3 | 6.2 |
| 1B | 1.69 | 70.1 | 46.2 | 27.5 | 14.1 | 10.6 | 7.0 |

Table 2: Columns under "% Savings" show the savings, in percentage, for the setting of Johnson et al. [2019] where the number of clusters is held fixed to approximately $\sqrt{n}$. Savings in bytes-per-element are shown in the second column, where $\log|\Pi| = \sqrt{n}\log((\sqrt{n}-1)!)$, and agree with Table 1. For external ids, 8 bytes are added to $\frac{1}{8n}\log|\Pi|$ to compute the total size per element, as well as to the cost under RCC. Meanwhile, $\log(n)$ bits are added to $\frac{1}{8n}\log|\Pi|$ for sequential ids, but not to the cost under RCC, as RCC does not require ids to represent clustering information. See Section 5.3 for a full discussion.

elements, which is the worst-case savings for both ROC and RCC. RCC achieves optimal savings for all combinations of datasets, number of elements, and clusters. ROC-2 has similar performance to RCC but requires significantly more compute as shown in Figure 2.

The total savings will depend on the cluster sizes, the number of bytes used to encode each element (i.e., FAISS vector/embedding), as well as the size of id numbers in the database. Cluster sizes are often set to $\sqrt{n}$ resulting in $\log|\Pi| = \sqrt{n}\log((\sqrt{n}-1)!)$ [Johnson et al., 2019]. A vast literature exists on encoding methods for vectors [Chen et al., 2010, Martinez et al., 2016, Babenko and Lempitsky, 2014, Jegou et al., 2010, Huijben et al., 2024] with typical values ranging from 8 to 16 bytes for BigANN and 4 to 8 for SIFT1M. Typically 8 bytes are used to store database ids when they come from an external source and have semantics beyond the vector database itself. Alternatively, ids are assigned sequentially taking up $\log(n)$ bits each when their only purpose is to be stored as sets to represent clustering information. These ids can be removed if vectors are stored with RCC as the clustering information is represented by the relative orderings between objects without the need for ids. Table 2 shows savings for RCC for the setting of Johnson et al. [2019] with $k \approx \sqrt{n}$ clusters.

# 6 Discussion

This work provides an efficient lossless coding algorithm for storing random clusters of objects from arbitrary sets. Our method, Random Cycle Coding (RCC), stores the clustering information in the ordering between encoded objects and does away with the need to assign meaningless labels for storage purposes.

A random clustering can be decomposed into 2 distinct mathematical quantities, the data set of objects present in the clusters (i.e., the union of all clusters), and an equivalence class representing the cluster assignments. For a given clustering, the equivalence class contains all possible orderings of the data that have cycles with the same elements as some cluster. The logarithm of the equivalence class size is exactly the number of bits needed to communicate an ordering of the data points, with the wanted permutation cycles, which we refer to as the *order information*. This quantity was previously defined by Varshney and Goyal [2006] as the amount of bits required to communicate an ordering of a sequence if the multiset of symbols was given, and equaled $\log n!$ when there are no repeated symbols. In the cluster case, the order information $\log|\Pi|$ is strictly less than $\log n!$ as the clustering carries partial information regarding the ordering between symbols in the following way: given the clustering, only orderings with the corresponding cycle structure will be communicated.

The savings achieved by RCC equals exactly the *assignment information* of the data, implying RCC is optimal in terms of compression rate for the probability model shown in Equation (8). The computational complexity of RCC scales with the number of bits recovered by bits-back, and reverts back to that of compressing a sequence when all clusters are atomic.

The savings for RCC scales quasi-linearly with the cluster sizes, and is independent of the representation size of the data. The experiments on real-world datasets from vector similarity search databases showcases where we think our method is most attractive: clusters of data requiring few bytes per element to communicate, where the bits-back savings can represent a significant share of the total representation size.

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

## A  Python code for RCC

```python
1  def element_codec_encode(state: int, element, precision: int) -> int:
2      ''' Encodes `element` into the ANS state using some codec '''
3      ... # definition of codec goes here
4      return next_state
5
6  def uniform_codec_decode(state: int, precision: int) -> (int, int):
7    index = state % precision
8    previous_state = state // precision
9    return index, previous_state
10
11 def encode(clustering: set[set], state: int) -> int:
12   clustering = sorted(map(sorted, clustering), reverse=True)
13   for cluster in reversed(clustering):
14     smallest_element = cluster.pop(0) # pop the first (smallest) element
15     while cluster: # Implement ROC: encode remaining elements as a set
16       precision = len(cluster)
17       index, state = uniform_codec_decode(state, precision)
18       element = cluster.pop(index)
19       state = element_codec_encode(element, state)
20     state = element_codec_encode(smallest_element, state)
21   return state
```

```python
1  import bisect
2
3  def symbol_codec_decode(state: int, element, precision: int):
4      ''' Decodes `element` from the ANS state using some codec '''
5      ... # definition of codec goes here
6      return element, previous_state
7
8  def uniform_codec_encode(state: int, index: int, precision: int) -> int:
9    next_state = state * precision + index
10   return next_state
11
12 def append_to_cluster_and_sort(cluster: list, element) -> list:
13   # Insert element into sorted list. The resulting list is sorted as well.
14   # This operation is in-place
15   bisect.insort(cluster, element)
16   return cluster
17
18 def decode(state: int, n: int) -> (list[list], int):
19   clustering = list()
20   smallest_element = float('inf')
21   while sum(map(len, clustering)) < n # loop until all elements are seen
22     element, state = symbol_codec_decode(state)
23     if element > smallest_element:
24       cluster = append_to_cluster_and_sort(cluster, element)
25       index = cluster.index(element) # get index of element in sorted cluster
26       state = uniform_codec_encode(state, index, precision=len(cluster))
27     else:
28       cluster = [element]
29       clustering.append(cluster)
30       smallest_element = element
31   return clustering, state
```

Figure 4: Pseudo-code for encoding (top) and decoding (bottom) a clustering of $n$ elements with our method, Random Cycle Coding (RCC). `uniform_codec_decode` samples an integer from `state` between $0$ and `precision`$-1$. `append_to_cluster_and_sort` inserts an element into a sorted list such that the resulting list is also sorted.

