# OpenReview forum: "Random Cycle Coding: Lossless Compression of Cluster Assignments via Bits-Back Coding"
_NeurIPS.cc/2024/Conference — NeurIPS 2024 poster_

### Official Review · Reviewer_MJQw · 2024-07-12

**Soundness:** 3
**Presentation:** 3
**Contribution:** 2
**Rating:** 5
**Confidence:** 2

**Summary:**

This paper proposes the random cycle coding that achieves optimal rate by bits-back coding techniques for cluster assignments. In addition, the newly proposed algorithm requires less computing resources, making it more practical.

**Strengths:**

For the cluster assignments, this paper proposes an optimal compression coding by bits-back coding techniques. In the coding view, the result is significant due to the optimal property. The RCC algorithm needs less computing and memory costs, making it more practical.

**Weaknesses:**

This paper borrows bits-back coding techniques and random order coding, making this paper quite incremental in the view of techniques. The difficulty in combing these two techniques is not clearly presented.

**Questions:**

This paper stresses the application in cluster assignments. Is there any other potential application? What is the special feature in cluster assignment that is compatible with RCC?

**Limitations:**

No. The limitations are not discussed explicitly.

---

> ### Author Rebuttal · Authors · 2024-08-06
>
> > This paper borrows bits-back coding techniques and random order coding, making this paper quite incremental in the view of techniques. The difficulty in combing these two techniques is not clearly presented.
>
> RCC is a very different algorithm from ROC.
> - ROC is a cluster-compression method, and was readapted to be used as baselines, as there are currently no baselines for optimal cluster compression.
> - ROC is the only existing optimal set compression algorithm, to the best of our knowledge, while RCC compresses arbitrary clusters, or partitions, of datapoints.
> - RCC efficiently maps each possible partitioning of datapoints to a subset of permutations with a common cycle structure. ROC does not in any way utilize the cycle structure of permutations to communicate information.
>
>
> The fact that RCC can re-use ROC is a feature and not a limitation of the contribution. A significant effort was made to understand and bridge both methods, so that RCC could be built on top of the efficient implementations available for ROC.
>
> > This paper stresses the application in cluster assignments. Is there any other potential application? What is the special feature in cluster assignment that is compatible with RCC?
>
> In its most general form, RCC is an algorithm for compressing random partitions of arbitrary sets (see [1] for background). The insight provided by this paper is that it is possible to map cluster assignments to the cycle structure of permutations, where each disjoint cycle defines one of the sets in the cluster.
>
> We will add an example to the appendix highlighting how cycles can be used to store the information of cluster membership for arbitrary data.
>
> [1] https://www.stat.uchicago.edu/~pmcc/pubs/partition10.pdf

---

> > ### Comment · Reviewer_MJQw · 2024-08-13
> > **Thank the authors for the rebuttal.**
> >
> > The rebuttal addresses my major concerns and I raise the score accordingly.

---

### Official Review · Reviewer_CCTj · 2024-07-12

**Soundness:** 3
**Presentation:** 3
**Contribution:** 3
**Rating:** 5
**Confidence:** 3

**Summary:**

## Summary
* This paper propose an entropy coding technique named RCC, which is the first one to achieve the optimal rate for cluster assignment. Theoretical and empirical results show that the rate saving and speed up of the proposed approach over previous suboptimal work ROC is evident when the number of cluster is close to sample size.

**Strengths:**

## Strength
* The paper presents a entropy coding technique that achieve optimal rate for cluster assignment, which is a step forward the previous work on multiset.

**Weaknesses:**

## Weakness
* Despite its theoretical optimality, the bitrate saving of RCC over ROC is a bit of marginal, especially when the number of clusters are not so large compared with number of elements. For a less extreme case when number of cluster is much smaller than number of elements (~\sqrt{n}), the difference between RCC and ROC can be quite small. (See Fig 3, n=10^7, cluster = 10^7 and cluster = 10^3.5).
* Similarly, the RCC approach has obviously better complexity compared with ROC, only when we have number of clusters close to number of samples. For the case when #cluster ~\sqrt{n}, the difference of temporal complexity can be less significant.

**Questions:**

## Questions
* What is the common practical #clusters compared with #samples? Is that common to have #clusters close to #samples?

**Limitations:**

Yes

---

> ### Author Rebuttal · Authors · 2024-08-06
>
> We thank the reviewer for the relevant practical questions.
>
> Most questions center around a comparison to ROC. Indeed there is a regime where the compression gain our method provides, RCC, is marginal over ROC (when $k$ is small). However, there are regimes where the gain is substantial (when $k$ is large). In all regimes, our method outperforms ROC while also being faster.
>
> Given that RCC provides both better compression rates and, guaranteedly, better wall-times, as well memory and computational complexities, the message we would like to highlight is: **in all regimes of $k$, there is no reason to use ROC over our method.**
>
> > Despite its theoretical optimality, the bitrate saving of RCC over ROC is a bit of marginal, especially when the number of clusters are not so large compared with number of elements. For a less extreme case when number of cluster is much smaller than number of elements (~\sqrt{n}), the difference between RCC and ROC can be quite small. (See Fig 3, n=10^7, cluster = 10^7 and cluster = 10^3.5).
>
> The performance of RCC is comparable to that of ROC-2, but RCC significantly outperforms ROC-1. In the regime of $k \approx \sqrt(n)$, where $k$ is the number of clusters, the RCC can provide gains of up to 20% with respect to ROC-1 on benchmark datasets such as SIFT1M.
>
> > Similarly, the RCC approach has obviously better complexity compared with ROC, only when we have number of clusters close to number of samples. For the case when #cluster ~\sqrt{n}, the difference of temporal complexity can be less significant.
>
> Indeed RCC is always better, both in terms of compression ratio and speed of execution, than ROC for any configuration of clusters. The difference in performance diminishes as the number of clusters decreases, and achieves equality at the extreme $k=n=1$ where all elements are in the same set/cluster. For the typical scenario of vector databases, where $k \approx \sqrt(n)$, RCC is at least 20% faster (according to the experiments in Figure 2) than ROC, while achieving the same or better compression ratio.
>
> > What is the common practical #clusters compared with #samples? Is that common to have #clusters close to #samples?
>
> In FAISS [1] the practical regime is $k = c \cdot \sqrt(n)$, where $c$ is some constant. The actual value will depend on the specific nature of the database and the algorithm used to perform vector compression.
>
> While this regime for $k$ is the desired one, in practice, there are distribution shifts that happen in production. Initially, the assignment algorithm, i.e., the algorithm that decides where a new vector should be placed in the database, is trained on some training set targeting $k = c \cdot \sqrt(n)$. During deployment, when a new vector needs to be added, if the real-world distribution changes, there is no guarantee that this target will be met. The number of clusters can drift over time to have significantly more, or less, cluster. RCC provides a way to robustly compress the database, irrespective of this distribution shift, while ROC does not.

---

### Official Review · Reviewer_rKbY · 2024-07-17

**Soundness:** 3
**Presentation:** 2
**Contribution:** 3
**Rating:** 6
**Confidence:** 2

**Summary:**

This paper study the problem of cluster assignments of datasets, the goal of which is encoding datasets along with their cluster information. The authors propose Random Cycle Coding method that utilizes cycle representation of permutation as indicators for cluster. The cycle is formed by a sequence of numbers and it represents a cluster of the data samples corresponding to the numbers in the cycle.
It is shown that the method outperforms other baseline methods while requires less computational complexity and memory.

**Strengths:**

The idea in employing the cycle representation of permutation to assign data to cluster is simple yet effective. The authors formalize the idea and show the effectiveness of their proposed method. The performance of the method is consistently outperforming others.

**Weaknesses:**

Although the main idea is simple to understand, it is quite hard to grasp the underlying techniques like ANS, ROC and bit-back coding from the paper. This paper compares their method only with ROC, which I think is not sufficient.

**Questions:**

1. What are the differences between RCC and ROC in cluster assignment problem? From my understanding, RCC stores cluster information by permutation. Then, how ROC-1 and ROC-2 store the clusterings?
2. Is ROC only existing baseline method?
3. To leveraging permutation, it seems sequential stacking is required, and parallel stacking or decoding is not allowed. If it is, are there any problems or disadvantages caused by not using parallel operations?
4. When extremely many data samples are stored, it seems that retrieving a cluster information for a data sample requires decoding every data sample if the sample is stacked first. Is this correct?

---

> ### Author Rebuttal · Authors · 2024-08-06
>
> > What are the differences between RCC and ROC in cluster assignment problem? From my understanding, RCC stores cluster information by permutation. Then, how ROC-1 and ROC-2 store the clusterings?
>
> That is correct, RCC stores the cluster information in the disjoint cycles of the permutation. ROC-1 and ROC-2 compress each cluster separately as a set of objects, so the clustering assignment is implicitly stored by knowing which set you are decoding.
>
> > Is ROC only existing baseline method?
>
> To the best of our knowledge, there is no existing method that addresses compression of clustering assignments. However, a cluster can be seen as a collection of disjoint sets. It is therefore natural to adapt set compression algorithms, such as ROC, to compress clusters.
>
> > To leveraging permutation, it seems sequential stacking is required, and parallel stacking or decoding is not allowed. If it is, are there any problems or disadvantages caused by not using parallel operations?
>
> This is correct: RCC does not allow “random access” to elements in the clusters. This is a limitation of all bits-back coding techniques. We will highlight this point in the camera-ready.
>
> > When extremely many data samples are stored, it seems that retrieving a cluster information for a data sample requires decoding every data sample if the sample is stacked first. Is this correct?
>
> This is partially correct. Since encoding requires randomizing the order, the element you are searching for can be in any position of the stack with equal probability. However, it is not necessary to decode the entire stack to retrieve the element, decoding can be stopped as soon as the element is found.

---

> ### Comment · Reviewer_rKbY · 2024-08-12
>
> Thank the authors for the rebuttal. The rebuttal answers my questions, and I have no further questions. Accordingly, I will raise my score.

---

### Official Review · Reviewer_BkEs · 2024-07-25

**Soundness:** 2
**Presentation:** 2
**Contribution:** 1
**Rating:** 5
**Confidence:** 3

**Summary:**

This paper proposes a coding scheme for lossless compression of cluster assignments. The proposed coding scheme is based on random order coding (ROC) with bits-back coding. Analysis and experiments show that it outperforms two variants of ROC in complexity and compression rate.

**Strengths:**

The presentation and research methodology presented in this paper are good, and a comparison with baseline methods is included. Examples are provided for clear explanation.

**Weaknesses:**

After carefully reading this work, I believe it is out of the scope of NeurIPS. This work applies the Random Order Coding (ROC) with bits-back coding for encoding clustering assignments. The main contribution is on the way to adapt the ROC to clustering assignment encoding, which does not include any discussion or consideration on the topics listed on NeurIPS 2024 Call For Paper.

Additionally, the paper does not provide any literature on clustering assignment compression. I am not a database researcher, but is the clustering assignment a significant overhead in practice? According to the author, "the number of bits used to store the vector embedding ranges from 4 to 16 bytes, while ids are typically stored as 8-byte integers" It is very unexpected that, in practice, the id takes up more than half of the storage. Could the author provide more references or elaborate more on this?

**Questions:**

See weakness

**Limitations:**

See weakness

---

> ### Author Rebuttal · Authors · 2024-08-06
>
> > After carefully reading this work, I believe it is out of the scope of NeurIPS. [...] does not include any discussion or consideration on the topics listed on NeurIPS 2024 Call For Paper.
>
> This work fits in the call for papers under “Infrastructure”, in the sub-category “improved implementation and scalability”. Furthermore, there have been many papers on this exact topic at NeurIPS, in the category of “Probabilistic methods”, as well as at sister conferences such as ICML and ICLR (see references).
>
> The target application of this compression algorithm is for vector databases, which are widely used in practice for embedding retrieval in foundation models, as well as similarity search, the most famous of which is FAISS [1]. Vector databases provide fast similarity search via efficient KNN search on accelerated hardware (e..g, GPUs). There is an extensive list of applications currently using vector databases such as for augmenting LLMs with memory (see [3] for an extensive discussion); as well as companies providing vector databases as a service such as Pinecone (https://www.pinecone.io/), Milvus (https://milvus.io/), Amazon Web Services (AWS, https://aws.amazon.com/what-is/vector-databases/), Zilliz (https://zilliz.com/), and many others.
>
> From a theoretical perspective, the main ideas of this paper advance the line of research on bits-back coding—an approach that was invented and has deep roots in the "probabilistic methods" community of NeurIPS; and adapts it to practical settings. See [4, 5, 6, 7, 8, 9, 10,11,12] for papers recently published at NeurIPS, as well as other venues, on bits-back coding
>
> Reducing the memory footprint of vector databases is important for many reasons, and has been highlighted in the official documentation of production-level vector databases [2, 3]. We highlight 2 of these reasons. First, it enables the use of better search algorithms in real-time which require more memory to run. Second, similarity search is usually done on batches of queries in practice; and, therefore, reducing the memory footprint of the database index enables us to increase the batch size and speed up the throughput of retrieval.
>
> - [1] https://github.com/facebookresearch/faiss
> - [2] https://github.com/facebookresearch/faiss/wiki/Indexes-that-do-not-fit-in-RAM
> - [3] (The Faiss library)[https://arxiv.org/abs/2401.08281]
> - [4] (ICLR 2019) [Practical Lossless Compression with Latent Variables using Bits Back Coding](https://arxiv.org/abs/1901.04866)
> - [5] (NeurIPS 2021) [Variational Diffusion Models](https://proceedings.neurips.cc/paper/2021/hash/b578f2a52a0229873fefc2a4b06377fa-Abstract.html)
> - [6] (NeurIPS 2021) [Maximum Likelihood Training of Score-Based Diffusion Models](https://proceedings.neurips.cc/paper/2021/hash/0a9fdbb17feb6ccb7ec405cfb85222c4-Abstract.html)
> - [7] (NeurIPS 2019) [Integer Discrete Flows and Lossless Compression](https://proceedings.neurips.cc/paper/2019/hash/9e9a30b74c49d07d8150c8c83b1ccf07-Abstract.html)
> - [8] (NeurIPS 2020) [Improving Inference for Neural Image Compression](https://proceedings.neurips.cc/paper/2020/hash/066f182b787111ed4cb65ed437f0855b-Abstract.html)
> - [9] (ICLR 2020) [HiLLoC: Lossless Image Compression with Hierarchical Latent Variable Models](https://arxiv.org/abs/1912.09953)
> - [10] (ICLR 2021) [IDF++: Analyzing and Improving Integer Discrete Flows for Lossless Compression](https://arxiv.org/abs/2006.12459)
> - [11] (NeurIPS 2019) [Compression with Flows via Local Bits-Back Coding](https://proceedings.neurips.cc/paper/2019/hash/f6e794a75c5d51de081dbefa224304f9-Abstract.html)
> - [12] (NeurIPS 2021) [iFlow: Numerically Invertible Flows for Efficient Lossless Compression via a Uniform Coder](https://proceedings.neurips.cc/paper/2021/hash/2e3d2c4f33a7a1f58bc6c81cacd21e9c-Abstract.html)
> - [13] (NeurIPS 2021) [Your Dataset is a Multiset and You Should Compress it Like One](https://openreview.net/forum?id=vjrsNCu8Km) - Best paper award NeurIPS 2021 Workshop on Deep Generative Models and Downstream Applications.
>
> > Additionally, the paper does not provide any literature on clustering assignment compression. I am not a database researcher, but is the clustering assignment a significant overhead in practice? [...] It is very unexpected that, in practice, the id takes up more than half of the storage. [...]?
>
> The size of the index is composed of the number of bits required to represent the ids plus the number of bits required to represent the vectors. Currently, there is a vast literature on lossy compression of vector databases (see [14] and [15] for a survey), which significantly reduces the size of these high-dimensional vectors to a few bytes. In many cases, the vectors can be reduced to less than 8 bytes per vector (note, not *per-dimension*, but 8 bytes to represent the entire vector) while still maintaining search performance useful for many applications (e.g., see Figure 3 of [14] where Deep1M is compressed to less than 8 bytes per vector). In extreme cases, such as [4], Table 2, the authors show that 4 bytes can be enough to maintain some level of recall performance.
>
> While lossy vector compression has been reasonably explored, very little to nothing has been done for the indices. Typically, users store custom indices together with the vectors (e.g., in FAISS this is done through the `index.add_with_ids` method where `index` is some inverted index). These indices can be of arbitrary size, and are commonly 32 bit (4 byte) or 64 bit (8 byte) integers. The indices therefore can represent a significant share of the memory required to store these databases.
>
> We will add a discussion to the introduction during the camera-ready to highlight that the lossy compression of vectors to less than 8 bytes increases the relevance of the index compression.
>
> - [14] (ICML 2024) [Residual Quantization with Implicit Neural Codebooks](https://arxiv.org/abs/2401.14732)
> - [15] https://github.com/erikbern/ann-benchmarks

---

> > ### Comment · Reviewer_BkEs · 2024-08-12
> >
> > Thanks for answering my questions. I have no further comments and questions. I will raise my rating.

---

### Official Review · Reviewer_h1ae · 2024-07-31

**Soundness:** 3
**Presentation:** 3
**Contribution:** 3
**Rating:** 6
**Confidence:** 3

**Summary:**

The paper introduces Random Cycle Coding (RCC), a method for lossless compression of cluster assignments in data sets. RCC encodes data sequentially, representing cluster assignments as cycles in permutations, which eliminates the need for artificial labels. The method is shown to achieve the Shannon bound in bit savings, and scales quasi-linearly with the largest cluster size. Empirical results show optimal byte savings across a range of cluster sizes for multiple datasets.

**Strengths:**

1. The proposed algorithm for encoding cluster assignments is novel and elegantly leverages the properties of clusterings and permutations to achieve theoretically optimal compression.

2. The approach also shows significant savings in experiments including on vector databases which can translate into gains across all machine learning approaches that rely on retrieval from vector databases.

**Weaknesses:**

Some of the technical details are not clearly explained (see question below) and overall, the details of the approach may be difficult to follow for audiences not familiar with the relevant source coding literature. I would recommend adding an example to illustrate how Algorithm 1 works end-to-end and why it works, either in the main paper or in the appendix, to remedy this.

**Questions:**

1. Why does the encode + decode time for RCC in Fig. 2 decrease as 'k' increases?

2. If the elements in each cycle are sorted (line 188) then how can the original order of the elements be recovered? If it cannot be recovered, then is this approach only limited to settings where the ordering of database elements is not important?

3. How is the centroid to cluster mapping information stored? I believe it will be needed to identify the clusters in which to search for the k nearest neighbors in the second stage of FAISS right?

---

> ### Author Rebuttal · Authors · 2024-08-06
>
> We thank the reviewer for the suggestion. An example of encoding and decoding a cluster assignment has been added to the appendix to further clarify the algorithm.
>
> > Why does the encode + decode time for RCC in Fig. 2 decrease as 'k' increases?
>
> This can be understood by noting that, in the experiment, as the number of clusters $k$ increases, the number of elements in each cluster $n_i = \frac{n}{k}$ decreases.
>
> The total complexity of RCC can be read from Algorithm 1, lines 2 and 3. The subroutine that encodes a single cluster of size $n_i$ (i.e., line 2 of Algorithm 1) has complexity $O(n_i \cdot \log n_i) = O(\frac{n}{k} \cdot \log \frac{n}{k})$. Line 3 encodes a single element, which is very fast, for every cluster, adding complexity $O(k)$. As $k$ increases, more and more time is spent on line 3 than line 2 of Algorithm 1.
>
> The total complexity of RCC is $O(k + n \cdot \log\frac{n}{k})$. If $k=1$, then all elements are in a single cluster, and the complexity is $O(n \cdot \log(n))$. At the other extreme, $k=n$, each element is in its own cluster, and the complexity is $O(n)$. In between these extremes, as $k$ increases, there will be more clusters (more time spent on line 3 of Algorithm 1), but every cluster will be smaller (less time spent on line 2 of Algorithm 1), and the latter is computationally more expensive.
>
> We will add this discussion to the camera-ready.
>
> > If the elements in each cycle are sorted (line 188) then how can the original order of the elements be recovered? If it cannot be recovered, then is this approach only limited to settings where the ordering of database elements is not important?
>
> This is correct: the order can never be recovered, but note this is by design. We assume clusters are sets of objects, i.e., unordered collections. The fact that elements have some order is due to the nature of how we represent information on a computer (i.e., memory is inherently sequential).
>
> This is the case for similarity search databases such as FAISS, for example, where the freedom to reorder vectors in a cluster/voronoi cell, as well as reordering the centroids (as long as it still aligns with the correct cluster).
>
> We will highlight this point in the camera-ready.
>
> > How is the centroid to cluster mapping information stored? I believe it will be needed to identify the clusters in which to search for the k nearest neighbors in the second stage of FAISS right?
>
> This is an important practical consideration, which is compatible with RCC. Line 1 in Algorithm 1 sorts the elements according to Foata’s Canonicalization (Definition 3.3). The centroids must be re-ordered to align with the respective centroid.
>
> We will add a discussion on what quantities we assume can be re-ordered to the introduction in the camera-ready.

---

> > ### Comment · Reviewer_h1ae · 2024-08-13
> > **Response to rebuttal**
> >
> > Thank you for addressing my concerns. As I had already recommended accepting the paper, I will keep my score.

---

### Decision · Program_Chairs · 2024-09-25

**Decision:**

Accept (poster)

**Comment:**

The paper considers the problem of compressing cluster assignments of arbitrary data sets. In other words, given that a clustering algorithm was run on a dataset and partitioned the datapoints into clusters, the proposed method compresses the clustering assignment information (the cluster labels). The paper proposes a compression algorithm called Random Cycle Coding (RCC), which encodes assignment information by viewing the clusters as cycles of a permutation (with respect to the natural ordering of the objects), and then "canonizes" the ordering within clusters and between the clusters. Experiments compare the performance of RCC with a similar method (but not specialized for clustering) called ROC. Practical gains over ROC are achieved when the number of clusters is large (e.g., ~sqrt{n}).

The reviewers agree that the proposed algorithm is elegant and interesting. RCC is faster than ROC and provides some compression gains (although only for very large values of k).